# Comparison of Long COVID-19 Caused by Different SARS-CoV-2 Strains: A Systematic Review and Meta-Analysis

**DOI:** 10.3390/ijerph192316010

**Published:** 2022-11-30

**Authors:** Min Du, Yirui Ma, Jie Deng, Min Liu, Jue Liu

**Affiliations:** 1Department of Epidemiology and Biostatistics, School of Public Health, Peking University, No. 38, Xueyuan Road, Haidian District, Beijing 100191, China; 2Institute for Global Health and Development, Peking University, No. 5, Yiheyuan Road, Haidian District, Beijing 100871, China; 3Global Center for Infectious Disease and Policy Research & Global Health and Infectious Diseases Group, Peking University, No. 38, Xueyuan Road, Haidian District, Beijing 100191, China; 4Key Laboratory of Reproductive Health, National Health and Family Planning Commission of the People’s Republic of China, No. 38, Xueyuan Road, Haidian District, Beijing 100191, China

**Keywords:** long COVID-19, long-term, variants, meta, review

## Abstract

Although many studies of long COVID-19 were reported, there was a lack of systematic research which assessed the differences of long COVID-19 in regard to what unique SARS-CoV-2 strains caused it. As such, this systematic review and meta-analysis aims to evaluate the characteristics of long COVID-19 that is caused by different SARS-CoV-2 strains. We systematically searched the PubMed, EMBASE, and ScienceDirect databases in order to find cohort studies of long COVID-19 as defined by the WHO (Geneva, Switzerland). The main outcomes were in determining the percentages of long COVID-19 among patients who were infected with different SARS-CoV-2 strains. Further, this study was registered in PROSPERO (CRD42022339964). A total of 51 studies with 33,573 patients was included, of which three studies possessed the Alpha and Delta variants, and five studies possessed the Omicron variant. The highest pooled estimate of long COVID-19 was found in the CT abnormalities (60.5%; 95% CI: 40.4%, 80.6%) for the wild-type strain; fatigue (66.1%; 95% CI: 42.2%, 89.9%) for the Alpha variant; and ≥1 general symptoms (28.4%; 95% CI: 7.9%, 49.0%) for the Omicron variant. The pooled estimates of ≥1 general symptoms (65.8%; 95% CI: 47.7%, 83.9%) and fatigue were the highest symptoms found among patients infected with the Alpha variant, followed by the wild-type strain, and then the Omicron variant. The pooled estimate of myalgia was highest among patients infected with the Omicron variant (11.7%; 95%: 8.3%, 15.1%), compared with those infected with the wild-type strain (9.4%; 95%: 6.3%, 12.5%). The pooled estimate of sleep difficulty was lowest among the patients infected with the Delta variant (2.5%; 95%: 0.2%, 4.9%) when compared with those infected with the wild-type strain (24.5%; 95%: 17.5%, 31.5%) and the Omicron variant (18.7%; 95%: 1.0%, 36.5%). The findings of this study suggest that there is no significant difference between long COVID-19 that has been caused by different strains, except in certain general symptoms (i.e., in the Alpha or Omicron variant) and in sleep difficulty (i.e., the wild-type strain). In the context of the ongoing COVID-19 pandemic and its emerging variants, directing more attention to long COVID-19 that is caused by unique strains, as well as implementing targeted intervention measures to address it are vital.

## 1. Introduction

As of 25 August 2022, a total of 596,119,505 confirmed cases and 6,457,101 deaths due to the coronavirus disease (COVID-19) have placed a huge disease burden, as well as delivered an economic loss, on the world [1]. It has now been more than two years since the COVID-19 pandemic was officially declared by the World Health Organization (WHO) on 11 March 2020. Further, more studies have now begun to direct attention to the long-term impact of COVID-19. Scientific questions on the long-term impact of COVID-19, specifically on matters such as the diagnosis, as well as in its definition, still require more research [2]. COVID-19 may have detrimental sequelae even after the post-acute phase, thereby depicting a new pathological condition: the “post-COVID-19 syndrome (PCS)” or “long COVID-19” [3]. Long COVID-19, also called post COVID-19 condition, was defined as a condition that occurs in individuals with a history of probable or confirmed severe acute respiratory syndrome coronavirus 2 (SARS-CoV-2) infection. The condition is usually 3 months from the onset of COVID-19 that incurred the symptoms, lasted for at least 2 months, and cannot be explained by an alternative diagnosis [4]. Except in cases of systemic symptoms, existing evidence has shown that long COVID-19 can also involve multiple systems, including the mental, nervous, respiratory, cardiovascular, digestive systems, etc. [5,6,7,8,9].

In the early stages of the pandemic, previous studies reported long COVID-19 that was caused by the wild-type strain. Further, a meta-analysis reported that the common symptoms of long COVID-19 caused by the wild-type strain were fatigue or muscle weakness, as well as mild dyspnea [10]. Since the beginning of the COVID-19 epidemic, SARS-CoV-2 has evolved, mutated, and produced variants with variance in transmissibility and virulence. The SARS-CoV-2 variants emerged from the original wild-type strain, which includes: Alpha (B.1.1.7), Beta (B.1.351), Gamma (P.1), Delta (B.1.617.2), and Omicron (B.1.1.529) [11]. These variants quickly became the main virus variants worldwide due to their increased transmissibility and virulence, a resistance to vaccine or acquired immunity from previous infection, and the ability to elude diagnostic detection [12].

Previous studies also reported on the differences in severity of COVID-19 cases via the different strains [13,14]. With the ongoing nature of COVID-19 and its emerging variants, the long COVID-19 of patients by unique variants should also be studied in order to avoid an excessive disease burden in the future. There have been a few studies that have reported on long COVID-19 during the different epidemic periods with the unique strains, but their results were consistent [15,16]. At present, there is no meta-analysis that has been conducted in order to provide evidence on the characteristics of long COVID-19 caused by the different strains. Thus, in this review, we aimed to systematically assess long COVID-19 that was caused by different SARS-CoV-2 strains at 3 months and above using the available evidence. This was performed so as to help better prepare management strategies and to better decease the long-term effects on infected people.

## 2. Methods

### 2.1. Search Strategy and Selection Criteria

The systematic literature review was reported in accordance with the preferred reporting items for systematic reviews and meta-analyses (PRISMA) checklist 2020. This review was also registered in PROSPERO (CRD42022339964).

We searched the PubMed, EMBASE, and ScienceDirect databases for studies without language restrictions, published up through 20 July 2022 with the following search terms: ((post COVID-19) OR (long COVID-19) OR ((COVID-19 OR SARS-CoV-2 OR coronavirus) AND ((long-term effect) OR sequelae OR (post condition) OR (post syndrome) OR (long-term consequence)))) AND (cohort OR (follow up) OR (case control study)). The initial searches were carried out by two investigators (MYR and DJ) independently.

The following PICO model was used to evaluate the study eligibility:(P)Participants: the long COVID-19 patients;(I)Intervention: the different SARS-CoV-2 strains;(C)Comparison: not applicable;(O)Outcome measures: the long COVID-19 effects and related issues, including clinical features (general symptoms, respiratory symptoms, cardiovascular symptoms, gastrointestinal symptoms, neurological, and psychiatric symptoms), pulmonary functional test (PFT), chest computerized tomography (CT]), and quality of life.

### 2.2. Inclusion and Exclusion Criteria

The inclusion criteria for selecting the cohort studies with reported confirmed dates or admissions, as well as the date or specific variants of long COVID-19 patients at 3 months and above were included. The following studies were excluded: (1) irrelevant to the subject of the meta-analysis, such as studies that did not use SARS-CoV-2 infection as the exposure; (2) insufficient data of long-term COVID-19 consequences; (3) duplicate studies or overlapping participants; (4) reviews, editorials, conference papers, case series/reports, secondary analysis, or animal experiments; and (5) qualitative designs. Studies were identified independently by two investigators (MYR and DJ) following the criteria above. This was achieved while discrepancies were solved by consensus or with a third investigator (DM).

### 2.3. Variants Identification and Data Extraction

In this study, we included studies which reported long COVID-19 that was caused by the wild-type strain and selected variants including the Alpha, Beta, Delta, Gamma, and Omicron strains (as defined in the terms of the WHO definitions [11]). If studies did not report specific strains, we then searched the genomic epidemiology of SARS-CoV-2 through the Global Initiative of Sharing All Influenza Data (GISAID) platform in order to find the main epidemic strain based on the period of the confirmed date or the admission date and country [17]. The results of the searches were screened in two stages. First, titles and abstracts were screened and, then, only were the relevant articles retained. Next, articles were read in detail—studies were selected for meta-analysis if they reported either results fitting our primary parameters (with CIs) or possessed sufficient information to facilitate the calculation of those values. The following data were extracted from the selected studies: (1) The basic information of the studies, including the first author, publication time, and country where the study was conducted; (2) the characteristics of the study population (including reported confirmed date or admission dates) or the specific strains of long COVID-19, as well as the sample size and follow-up period; (3) the clinical features of long COVID-19, including the number of cases with general long COVID-19 symptoms, i.e., respiratory symptoms, cardiovascular symptoms, gastrointestinal symptoms, neurological symptoms, and psychiatric symptoms, as well as the results of a PFT and CT; (4) the number of cases with problems in the five dimensions of the European Quality of Life Five-Dimension Five-Level Scale (EQ-5D-5L)—which is an instrument that was developed, by the EuroQol Group in 1987, for the purposes of describing and valuing health-related quality of life matters [18]. The data extraction and determination of information eligibility were independently conducted by the two investigators (MYR and DJ) following the criteria above, while discrepancies were solved by consensus or with a third investigator (DM).

### 2.4. Quality Assessment and Risk of Bias

We used the Newcastle–Ottawa quality assessment scale in order to evaluate the risk of bias in the included studies. The cohort studies were classified as having a low (≥7 stars), moderate (5–6 stars), or high risk of bias (≤4 stars), with an overall quality score of 9 stars. Quality assessment was independently conducted by two investigators (MYR and DJ), while discrepancies were solved by consensus or with a third investigator (DM).

### 2.5. Statistical Analysis

The Der Simonian and Laird method was performed in order to pool the prevalence (PP), as well as the 95% confidence interval (CI) of long COVID-19 at 3 months and above. Considering the fact that the heterogeneity was larger than 50% between the studies, random-effects models [19] were used in order to calculate the pooled effect and its 95% confidence interval (CI). In addition, publication bias was assessed by the Egger regression test [20]. After extracting all essential data using Excel 2021 (Microsoft Corporation), the data analyses were completed by using Stata 16.0. Moreover, two-sided *p* < 0.05 indicated statistical significance.

## 3. Results

### 3.1. Basic Characteristics

We identified 4710 records through PubMed, EMBASE, and ScienceDirect database searches. A total of 51 articles (i.e., 33,573 patients) were selected for analysis based on the inclusion and exclusion criteria [6,15,16,18,21,22,23,24,25,26,27,28,29,30,31,32,33,34,35,36,37,38,39,40,41,42,43,44,45,46,47,48,49,50,51,52,53,54,55,56,57,58,59,60,61,62,63,64,65,66,67]. The study selection process was documented in Figure 1. Over the 51 studies, the quality assessment—which was conducted in order to evaluate the risk of bias—for 45 of the studies and the 6 remaining found the quality to be low and moderate, respectively. Most were conducted in Spain (9), followed by Italy (7), and then China (6). A total of 38 studies only included patients infected with the wild-type strain, three with the Alpha variant, three with the Delta variant, and five with the Omicron variant. The other two were the Beta and Gamma variants (Appendix A).

### 3.2. Pooled Estimates of Long COVID-19

We included general symptoms; respiratory symptoms; cardiovascular symptoms; gastrointestinal symptoms; neurological and psychiatric symptoms; PFT; CT results; and quality of life evaluation. The top five symptoms with high pooled estimates were CT abnormalities (60.5%; 95% CI: 40.4%, 80.6%;); carbon monoxide diffusing capacity (DLCO) <80% (56.9%; 95% CI: 40.6%, 73.1%); ≥1 respiratory symptoms (51.1%; 95% CI: 41.4%, 60.8%); modified medical research council dyspnea scale (mMRC) = 0 (50.6%, 95% CI: 28.0%, 73.2%); and pain or discomfort (48.6%, 95% CI: 35.4%, 61.7%). In addition, there were 30 pooled estimates that were more than 10.0% in symptoms such as fatigue, cough, sleep difficulty, etc. The pooled estimates of the other symptoms, as well as the publication bias assessment via the Egger regression test are detailed in Appendix A.

### 3.3. Pooled Estimates of Long COVID-19 by Different Strains

In regard to the wild-type strain, the highest pooled estimate of long COVID-19 was found in CT abnormalities (60.5%; 95% CI: 40.4%, 80.6%) for 11 of the studies. In regard to the Alpha variant, the pooled estimates of ≥1 general symptoms, fatigue, cough and dyspnea were 65.8% (95% CI: 47.7%, 83.9%) for three studies, 66.1% (95% CI: 42.2%, 89.9%) for three studies, 34.2% (95% CI: 8.3%, 60.1%) for two studies, and 23.7% (95% CI: 2.0%, 45.5%) for two studies. Two studies with the Delta variant only gave the pooled estimates of sleep difficulty (2.5%; 95% CI: 0.2%, 4.9%). In regard to the Omicron variant, the top three estimates were ≥1 symptoms (931/4860; 28.4%; 95% CI: 7.9%, 49.0%), sleep difficulty (3082/16211; 18.7%; 95% CI: 1.0%, 36.5%), and fatigue (3457/15848; 18.1%; 95% CI: 0.4%, 35.8%). The specific results are shown in Table 1.

As shown in Table 1, the pooled estimates of ≥1 general symptoms and fatigue were highest among the patients infected with the Alpha variant, followed by the wild-type strain and the Omicron variant. The pooled estimate of myalgia was higher among patients infected with the Omicron variant (11.7%; 95%: 8.3%, 15.1%), compared with patients infected with the wild-type strain (9.4%; 95%: 6.3%, 12.5%). The pooled estimate of sleep difficulty was lowest among patients infected with the Delta variant (2.5%; 95%: 0.2%, 4.9%), compared with patients infected with the wild-type strain (24.5%; 95%: 17.5%, 31.5%) and the Omicron variant (18.7%; 95%: 1.0%, 36.5%).

## 4. Discussion

The emerging unique variants of SARS-CoV-2 indicate that long COVID-19 may be different due to changes in virulence between the variants. This systematic review and meta-analysis comprehensively assessed the long COVID-19 caused by different SARS-CoV-2 strains in order to supplement limited evidence. Generally, this study reported the most common long COVID-19 and its differences between the unique strains. Our findings showed that CT abnormalities (60.5%), carbon monoxide diffusing capacity <80% (56.9%), and ≥1 respiratory symptoms (51.1%) were common sequelae among COVID-19 patients, which are similar results to our previous study [10]. Groff et al. also reported that the most prevalent pulmonary sequelae was found in chest imaging abnormality (62.2%, 95%CI: 45.8–76.5%) [68]. Furthermore, SARS-CoV-2 infection generally causes direct and indirect injury of the pulmonary system by invasion and via a cytokine storm [69,70,71]. Our results indicated that the damage to the respiratory symptoms caused by SARS-CoV-2 may be prolonged. Therefore, a consideration on the persistent rehabilitation treatment that will be utilized is vital.

Importantly, our study first found, using meta-analysis, that long COVID-19 may be different according to what strain caused it. The Alpha variant was first reported in the United Kingdom in September 2020. Then, the Delta variant was reported in India in October 2020. Finally, the Omicron variant was reported in multiple countries in November 2021 [11]. In regard to the Delta variant, its infected wave was attributed to the epidemic across 98 countries in 2021, such as India, the UK, the US, and certain Southeast Asian countries [13]. According to the WHO report—as of epidemiological week 35—a total of 48 countries have reported detection of Omicron [72]. In respect of considering the rehabilitation of patients by the different strains in the future, understanding the differences of sequelae is crucial. Duong et al. reported that the Omicron variant was around one to three times the daily number of cases of hospitalization that is incurred in comparison with the Delta variant [13]. The Omicron variant has high transmissibility and severity. As such, the sequelae caused by it should not be ignored. Moreover, we found that the pooled estimate of sleep difficulty was low among patients infected with the Delta variant, but was high among patients infected with the wild-type strain and the Omicron variant. Meanwhile, the pooled estimate of myalgia was higher among patients infected with the Omicron variant, compared with patients infected with the wild-type strain. However, we found that the pooled estimates of ≥1 general symptoms and fatigue were more than sixty percent among patients infected with the Alpha variant, which was higher than those infected with the wild-type strain and the Omicron variant.

Close attention should be paid to the fact that vaccination may have had influences on our results. Duong et al. reported that the Omicron variant had less effect on the number of daily intensive care unit (ICU) cases, most likely due to the total number of vaccinated people in each country [13]. Yu et al. also found that the pooled proportion of asymptomatic infection and non-severe disease with Omicron were 25.5% and 97.9%, respectively, which is significantly higher than those of Delta with 8.4% and 91.4% [14]. The vaccination of a booster dose that may cause variants had less effect on the severity of COVID-19 or long COVID-19 when compared with the epidemic of the wild-type strain or the Alpha variant [73,74,75,76].

There is currently a lack of cohort studies that can be utilized to evaluate the CT and PFT results of long COVID-19 caused by unique variants. Thus, more research studies are needed. This study provided a part of the evidence that is needed for a full meta-analysis of long COVID-19 via its different strains.

Our study also possessed certain limitations. Although the identification of strains was based on the largest genomic epidemiology of the SARS-CoV-2 platform, GISAID, it is possible to have misclassified strains [17]. Therefore, our results should be interpreted with caution. Secondly, the incidence and evolution of long COVID-19 are dependent on complicated factors, including vaccination status and geographic region [68]. Due to the fact that the studies with variants that were possible to be included were limited, we could not, thus, analyze the effect of other factors on our results.

## 5. Conclusions

Although variants such as the Alpha variant are currently only prevalent in a few countries, the Delta and Omicron variants have become the dominant strains in many countries around the world [77]. Identifying the long COVID-19 of different strains is a key factor in designing an appropriate health management strategy. We found that the pooled estimates of ≥1 general symptoms and fatigue were more than sixty percent among patients infected with the Alpha variant, which was higher than those infected with the wild-type strain and the Omicron variant. Meanwhile, the pooled estimate of myalgia was higher among patients infected with the Omicron variant, compared with patients infected with the wild-type strain. The pooled estimate of sleep difficulty was low among patients infected with the Delta variant, but it was high among patients infected with the wild-type strain and the Omicron variant. In conclusion, our findings suggest that different strains could all cause long COVID-19. Furthermore, long COVID-19 that is caused by different strains has no differences, except in certain symptoms (including general symptoms and sleep difficulty that were found to be discrepant between the unique strains). In regard to the ongoing COVID-19 pandemic and its emerging variants, more attention needs to be directed to instances of long COVID-19 that are caused by unique strains. Furthermore, implementing targeted intervention measures to address them are vital.

## Figures and Tables

**Figure 1 ijerph-19-16010-f001:**
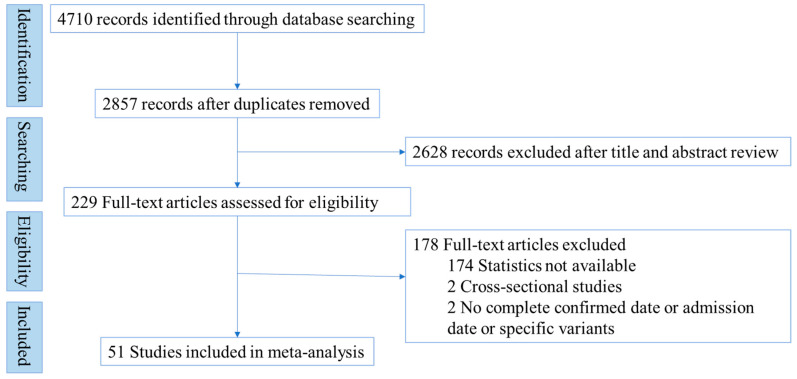
Study flow diagram.

**Table 1 ijerph-19-16010-t001:** Pooled prevalence of long COVID-19 by different strains at 3 months follow-up and above.

Consequences	Number of Studies	Patients n/N	PP (95% CI, %)	*p*-Value	I^2^
**General Symptoms**					
**≥1 symptoms**					
Alpha	3	475/779	65.8 (47.7, 83.9)	<0.05	96.8%
Delta	1	56/162	34.6 (27.2, 41.9)	<0.05	-
Omicron	2	931/4860	28.4 (7.9, 49.0)	<0.05	97.5%
Wild-type	18	3659/7069	52.1 (44.0, 60.1)	<0.05	98.0%
**Fever or Feverishness**					
Alpha	1	10/324	3.1 (1.2, 5.0)	<0.05	-
Omicron	3	178/11,535	8.0 (−1.7, 17.7)	>0.05	95.2%
Wild-type	9	101/3681	2.6 (1.3, 3.8)	<0.05	90.9%
**Fatigue**					
Alpha	3	494/760	66.1 (42.2, 89.9)	<0.05	98.1%
Beta	1	295/2198	13.4 (12.0, 14.8)	<0.05	-
Delta	1	40/162	24.7 (18.1, 31.3)	<0.05	-
Omicron	2	3457/15,848	18.1 (0.4, 35.8)	<0.05	99.9%
Wild-type	19	1234/6094	26.3 (20.7, 31.9)	<0.05	97.3%
**Muscle weakness**					
Delta	1	42/1354	3.1 (2.2, 4.0)	<0.05	-
Omicron	1	30/245	12.2 (8.1, 16.3)	<0.05	-
Wild-type	4	70/1730	4.2 (1.5, 6.9)	<0.05	87.7%
**Myalgia**					
Alpha	1	74/324	22.8 (18.3, 27.4)	<0.05	-
Delta	1	69/1354	5.1 (3.9, 6.3)	<0.05	-
Omicron	3	1346/11,539	11.7 (8.3, 15.1)	<0.05	64.8%
Wild-type	11	235/2727	9.4 (6.3, 12.5)	<0.05	96.0%
**Joint pain or arthralgia**					
Alpha	1	53/301	17.6 (13.3, 21.9)	<0.05	-
Omicron	4	2124/11,732	14.9 (8.3, 21.4)	<0.05	93.4%
Wild-type	4	152/602	24.3 (6.6, 42.0)	<0.05	95.8%
**Headache**					
Alpha	1	45/324	65.8 (47.7, 83.9)	<0.05	-
Beta	1	8/2198	34.6 (27.2, 41.9)	<0.05	-
Delta	5	32/1516	28.4 (7.9, 49.0)	<0.05	82.1%
Omicron	5	3276/16,157	52.1 (44.0, 60.1)	<0.05	99.9%
Wild-type	1	624/7604	10.0 (7.6, 12.4)	<0.05	95.8%
**Dizziness or Vertigo**					
Delta	1	1/162	0.6 (−0.6, 1.8)	>0.05	-
Omicron	3	2023/6074	1.6 (−0.8, 4.0)	>0.05	90.1%
Wild-type	6	175/3098	5.9 (2.7, 9.2)	<0.05	96.7%
**Olfactory abnormalities**					
Omicron	4	2651/16,123	10.2 (−3.2, 23.6)	>0.05	99.8%
Wild-type	4	46/773	7.0 (2.7, 11.4)	<0.05	84.7%
**Olfactory loss**					
Delta	1	23/1354	1.7 (1.0, 2.4)	<0.05	-
Omicron	1	62/222	27.9 (22.0, 33.8)	<0.05	-
Wild-type	9	418/3725	13.1 (8.5, 17.8)	<0.05	95.1%
**Taste abnormalities**					
Omicron	4	2030/16,090	8.7 (−2.3, 19.7)	>0.05	99.8%
Wild-type	3	40/703	7.1 (1.0, 13.2)	<0.05	90.9%
**Taste loss**					
Alpha	1	21/301	7.0 (4.1, 9.9)	<0.05	-
Beta	1	12/2198	0.5 (0.2, 0.9)	<0.05	-
Delta	1	34/1354	2.5 (1.7, 3.3)	<0.05	-
Omicron	1	50/222	22.5 (17.0, 28.0)	<0.05	-
Wild-type	7	328/3515	10.4 (6.1, 14.6)	<0.05	94.4%
**Hair loss**					
Omicron	2	1572/11,019	18.2 (8.2, 28.2)	<0.05	76.6%
Wild-type	7	209/3018	6.8 (3.4, 10.1)	<0.05	94.2%
**Cutaneous or Skin disorders**					
Wild-type	6	156/3738	3.9 (2.1, 5.7)	<0.05	88.3%
Rash	5	666/16,548	3.3 (0.4, 6.1)	<0.05	99.3%
Omicron	4	665/16,204	4.2 (0.3, 8.1)	<0.05	99.5%
Wild-type	1	1/344	0.3 (-0.3, 0.9)	>0.05	-
**Respiratory symptoms**					
**Cough**					
Alpha	2	87/452	23.7 (2.0, 45.5)	<0.05	95.7%
Delta	1	3/162	1.9 (−0.2, 3.9)	>0.05	-
Gamma	1	34/156	21.8 (15.3, 28.3)	<0.05	-
Omicron	2	1470/15,768	6.8 (−5.1, 18.7)	>0.05	99.9%
Wild-type	21	853/7691	13.4 (10.4, 16.5)	<0.05	97.4%
**Dyspnea**					
Alpha	2	125/429	34.2 (8.3, 60.1)	<0.05	96.4%
Beta	1	37/2197	1.7 (1.1, 2.2)	<0.05	-
Delta	1	14/1354	1.0 (0.5, 1.6)	<0.05	-
Gamma	1	68/158	43.0 (35.3, 50.8)	<0.05	-
Omicron	2	177/4860	9.2 (−3.3, 21.6)	>0.05	96.3%
Wild-type	20	1684/7469	23.3 (16.2, 30.5)	<0.05	98.9%
**Expectoration**					
Wild-type	5	80/998	7.5 (2.9, 12.1)	<0.05	94.8%
**Nasal congestion**					
Alpha	1	52/324	16.0 (12.1, 20.0)	<0.05	-
Omicron	1	18/4638	0.4 (0.2, 0.6)	<0.05	-
Wild-type	5	35/1049	2.8 (0.4, 5.1)	<0.05	85.5%
**Sore throat**					
Alpha	1	19/324	5.9 (3.3, 8.4)	<0.05	-
Delta	1	9/1354	0.7 (0.2, 1.1)	<0.05	-
Omicron	2	991/15,846	4.5 (−3.7, 12.7)	>0.05	99.9%
Wild-type	3	26/635	4.3 (0.9, 7.8)	<0.05	73.0%
**mMRC = 0**					
Alpha	1	97/312	31.1 (26.0, 36.2)	<0.05	-
Wild-type	5	702/1005	54.7 (33.9, 75.5)	<0.05	98.3%
**mMRC ≥ 1**					
Alpha	1	215/312	68.9 (63.8, 74.0)	<0.05	-
Omicron	1	25/73	34.2 (23.4, 45.1)	<0.05	-
Wild-type	6	350/1125	44.1 (26.3, 62.0)	<0.05	97.9%
**Cardiovascular symptoms**					
**Short Breath**					
Alpha	1	175/327	53.5 (48.1, 58.9)	<0.05	-
Delta	1	14/162	8.6 (4.3, 13.0)	<0.05	-
Omicron	1	3444/11,183	30.8 (29.9, 31.7)	<0.05	-
Wild-type	5	215/1530	17.6 (9.5, 25.8)	<0.05	96.5%
**Palpitations**					
Delta	1	7/1354	0.5 (0.1, 0.9)	<0.05	96.8%
Omicron	3	87/4956	5.9 (1.1, 10.7)	<0.05	-
Wild-type	9	191/3454	5.7 (3.2, 8.3)	<0.05	97.5%
**Gastrointestinal symptoms**					
**≥1 symptoms**					
Wild-type	5	35/1603	1.8 (0.2, 3.3)	<0.05	84.3%
**Loss of appetite**					
Delta	2	53/1516	2.2 (−0.9, 5.4)	>0.05	93.7%
Omicron	2	26/318	11.9 (−3.8, 27.6)	>0.05	90.6%
Wild-type	4	28/876	3.0 (0.7, 5.3)	<0.05	74.6%
**Nausea**					
Delta	1	1/162	0.6 (−0.6, 1.8)	>0.05	-
Omicron	2	505/11,353	15.3 (−7.3, 37.9)	>0.05	94.9%
Wild-type	3	14/710	1.5 (−0.5, 3.5)	>0.05	75.9%
**Diarrhea**					
Omicron	2	505/11,353	4.4 (−1.5, 10.2)	>0.05	70.8%
Wild-type	8	14/710	2.5 (1.0, 3.9)	<0.05	92.5%
**Abdominal pain**					
Omicron	3	636/16,163	3.2 (−1.1, 7.5)	>0.05	99.7%
Wild-type	6	58/2164	2.0 (0.5, 3.5)	<0.05	87.4%
**Constipation**					
Delta	1	35/1354	2.6 (1.7, 3.4)	<0.05	-
Omicron	1	26/245	10.6 (6.8, 14.5)	<0.05	-
Wild-type	3	38/1370	3.1 (0.4, 5.8)	<0.05	92.2%
**Neurological and psychiatric symptoms**					
≥1 symptoms					
Wild-type	8	371/2950	13.8 (8.5, 19.2)	<0.05	95.7%
**Paresthesias**					
Delta	1	6/1354	0.4 (0.1, 0.8)	<0.05	-
Omicron	1	1337/11,236	11.9 (11.3, 12.5)	<0.05	-
Wild-type	7	161/1773	12.7 (7.7, 17.7)	<0.05	95.1%
**Memory problem**					
Omicron	1	1794/11,174	16.1 (15.4, 16.7)	<0.05	-
Wild-type	6	225/1489	17.3 (8.7, 25.9)	<0.05	97.2%
**Sleep difficulty**					
Alpha	1	151/327	46.2 (40.8, 51.6)	<0.05	-
Delta	2	51/1516	2.5 (0.2, 4.9)	<0.05	82.2%
Omicron	4	3082/16,211	18.7 (1.0, 36.5)	<0.05	99.9%
Wild-type	11	474/3067	24.5 (17.5, 31.5)	<0.05	98.5%
**Depression**			**1**		
Delta	1	1/162	0.6 (−0.6, 1.8)	>0.05	-
Omicron	1	2274/11,149	20.4 (19.6, 21.1)	<0.05	-
Wild-type	9	411/3585	19.7 (10.1, 29.4)	<0.05	99.1%
**Anxiety**					
Delta	1	1/162	0.6 (−0.6, 1.8)	>0.05	-
Omicron	1	1196/11,174	10.7 (10.1, 11.3)	<0.05	-
Wild-type	11	442/3134	15.3 (9.7, 20.8)	<0.05	96.8%
**Difficulty concentrating**					
Omicron	2	3542/15,817	16.0 (−14.5, 46.4)	>0.05	100.0%
Wild-type	6	370/2276	23.3 (14.9, 31.6)	<0.05	96.1%
**PFT**					
**FEV1 < 80%**					
Alpha	1	31/123	25.2 (17.5, 32.9)	<0.05	-
Wild-type	4	39/302	13.4 (7.0, 19.9)	<0.05	62.9%
**TLC < 80%**					
Alpha	1	30/111	27.0 (18.8, 35.3)	<0.05	-
Delta	1	76/121	62.8 (54.2, 71.4)	<0.05	-
Wild-type	4	98/596	17.2 (8.6, 25.8)	<0.05	87.8%
**DLCO < 80%**					
Alpha	1	50/111	45.0 (35.8, 54.3)	<0.05	-
Wild-type	5	816/1302	59.2 (40.4, 78.1)	<0.05	96.7%
**CT results**					
**CT abnormalities**					
Wild-type	11	1540/2206	60.5 (40.4, 80.6)	<0.05	99.2%
**GGO**					
Wild-type	12	403/1231	38.9 (26.8, 51.0)	<0.05	95.7%
**Consolidation**					
Wild-type	5	21/388	5.4 (0.8, 9.9)	<0.05	80.6%
**Fibrosis**					
Wild-type	9	340/1790	24.4 (13.3, 35.4)	<0.05	96.3%
**Bronchiectasis**					
Wild-type	7	195/892	22.2 (9.7, 34.7)	<0.05	96.6%
**EQ-5D-5L**					
**Mobility**					
Alpha	1	65/128	50.8 (42.1, 59.4)	<0.05	-
Omicron	1	28/73	38.4 (27.2, 49.5)	<0.05	-
Wild-type	3	99/837	11.7 (9.5, 13.9)	<0.05	0.0%
**Personal care**					
Omicron	1	28/73	68.5 (57.8, 79.1)	<0.05	-
Wild-type	3	99/837	2.8 (−0.3, 6.0)	>0.05	88.5%
**Usual activity**					
Alpha	1	69/128	53.9 (45.3, 62.5)	<0.05	-
Omicron	1	37/73	50.7 (39.2, 62.2)	<0.05	-
Wild-type	2	79/595	15.2 (5.2, 25.1)	<0.05	88.2%
**Pain or discomfort**					
Alpha	1	78/128	60.9 (52.5, 69.4)	<0.05	-
Omicron	1	52/73	71.2 (60.8, 81.6)	<0.05	-
Wild-type	3	300/838	37.2 (30.1, 44.3)	<0.05	76.9%
**Anxiety and depression**					
Omicron	1	53/73	72.6 (62.4, 82.8)	<0.05	-
Wild-type	4	724/941	28.0 (7.0, 49.0)	<0.05	98.8%

Abbreviations—mMRC: Modified medical research council dyspnea scale; PFT: pulmonary functional test; FEV1: forced expiratory volume in one second; TLC: total lung capacity; DLCO: carbon monoxide diffusing capacity; CT: computerized tomography; GGO: ground-glass opacity; and EQ-5D-5L: quality of life evaluation.

## Data Availability

Data are available from the corresponding authors on request.

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
