# Peer review of "Comparison of Long COVID-19 Caused by Different SARS-CoV-2 Strains: A Systematic Review and Meta-Analysis"

_ijerph, 2022, doi:10.3390/ijerph192316010_

Round 1

Reviewer 1 Report

17-18 the idea is clear, but the statement is difficult to read ... reformulated

21 I suggest evaluation via the Joanna Briggs ..

I think it not necessary to describe how the meta analysis was conducted .. for the PRISMA guidelines describe Population, Intervention, Comparator and Outcome.

Make the results discursive, objectify whether any symptom of the syndrome is more associated with one variant or another.

38 Long COVID

46  I suggest a statement like this: << However, COVID-19 might have detrimental sequelae even after the post-acute phase, depicting a new pathological condition: the “post-COVID-19 syndrome (PCS)” or “long COVID” >> reference: https://www.mdpi.com/2076-3417/12/17/8593

I recommend putting PROSPERO in the abstract to increase the caliber of the manuscript

92 I would recall the PICO, above all I would underline how you have divided the variants with bibliographic references .. what does the Wild variant refer to? Are you sure your definition is globally accepted?

121 I suggest to write that heterogeneity is taken for granted

Figure 2 Not scientifically appropriate, to remove ..

Author Response

Reviewer: 1

17-18 the idea is clear, but the statement is difficult to read ... reformulated

Response: Thanks. We have restated this sentence as shown in page 1 line 17-19: “Although several studies were conducted to review the data of long COVID-19, research systematically assessing differences of long COVID-19 caused by unique SARS-CoV-2 variant was lacking” changed as “Although many studies of long COVID-19 were reported, systematical research assessing differences of long COVID-19 caused by unique SARS-CoV-2 strains was lack.”.

21 I suggest evaluation via the Joanna Briggs .

Response: Thanks for your suggestions. Considering the assessment tool related to COVID-19 was Newcastle–Ottawa quality assessment scale (Deng J, Ma Y, Liu Q, Du M, Liu M, Liu J. Comparison of the Effectiveness and Safety of Heterologous Booster Doses with Homologous Booster Doses for SARS-CoV-2 Vaccines: A Systematic Review and Meta-Analysis. Int J Environ Res Public Health. 2022 Aug 29;19(17):10752. doi: 10.3390/ijerph191710752. PMID: 36078466; PMCID: PMC951778; Ma Y, Deng J, Liu Q, Du M, Liu M, Liu J. Long-Term Consequences of COVID-19 at 6 Months and Above: A Systematic Review and Meta-Analysis. Int J Environ Res Public Health. 2022;19(11):6865. Published 2022 Jun 3. doi:10.3390/ijerph19116865; Notarte KI, Catahay JA, Velasco JV, et al. Impact of COVID-19 vaccination on the risk of developing long-COVID and on existing long-COVID symptoms: A systematic review. EClinicalMedicine. 2022;53:101624. Published 2022 Aug 27. doi:10.1016/j.eclinm.2022.10162), in order to ensure comparability with previous studies, we used the common evaluation tool.

I think it not necessary to describe how the meta analysis was conducted .. for the PRISMA guidelines describe Population, Intervention, Comparator and Outcome.

Response: Thanks. We have described it using PICO in abstract as shown in page 1 line 23-26: “We systematically searched PubMed, EMBASE, and ScienceDirect to find cohort studies of long COVID-19 defined by WHO. The main outcomes were the percentages of long COVID-19 among patients infected with different SARS CoV-2 strains.” and added it in methods as shown in page 3 line 102-109: “The following PICO model was used to evaluate the study eligibility:

(P) Participants: long COVID-19 patients;

(I) Intervention: different SARS-CoV-2 strains;

(C) Comparator: not applicable;

(O) Outcome measure: long COVID-19 including clinical features (general symptoms, respiratory symptoms, cardiovascular symptoms, gastrointestinal symptoms, neurological and psychiatric symptoms), pulmonary functional test (PFT), chest computerized tomography (CT]) and quality of life.”

Make the results discursive, objectify whether any symptom of the syndrome is more associated with one variant or another.

Response: Thanks. We have restated results as shown in page 1 line 31-41:” The highest pooled estimate of long COVID-19 was CT abnormalities (60.5%; 95% CI: 40.4%, 80.6%) for wild-type strain; fatigue (66.1%; 95% CI: 42.2%, 89.9%) for Alpha variant; ≥1 general symptoms (28.4%; 95% CI: 7.9%, 49.0%) for Omicron variant. The pooled estimates of ≥1 general symptoms (65.8%; 95% CI: 47.7%, 83.9%) and fatigue were highest among patients infected with Alpha variant, followed by wild-type strain and Omicron variant. The pooled estimate of myalgia was higher among patients infected with Omicron variant (11.7%; 95%: 8.3%, 15.1%), compared with those infected with wild-type strain (9.4%; 95%: 6.3%, 12.5%). The pooled estimate of sleep difficulty was lowest among patients infected with Delta variant (2.5%; 95%: 0.2%, 4.9%), compared with those infected with wild-type strain (24.5%; 95%: 17.5%, 31.5%) and Omicron variant (18.7%; 95%: 1.0%, 36.5%).”.

38 Long COVID

Response: Thanks. We have changed keywords “COVID-19” as Long COVID-19”

46  I suggest a statement like this: << However, COVID-19 might have detrimental sequelae even after the post-acute phase, depicting a new pathological condition: the “post-COVID-19 syndrome (PCS)” or “long COVID” >> reference: https://www.mdpi.com/2076-3417/12/17/8593

Response: Thanks. We used the definition from WHO, in addition, we have added this statement as shown in page 2 line 57-59 :” COVID-19 might have detrimental sequelae even after the post-acute phase, depicting a new pathological condition: the “post-COVID-19 syndrome (PCS)” or “long COVID -19” [3]”.

I recommend putting PROSPERO in the abstract to increase the caliber of the manuscript

Response: Thanks. We have added PROSPERO in the abstract as shown in page 1 line 26-27:” This study was registered in PROSPERO (CRD42022339964).”

92 I would recall the PICO, above all I would underline how you have divided the variants with bibliographic references .. what does the Wild variant refer to? Are you sure your definition is globally accepted?

Response: Thanks. In our study, we included wild-type strain, and variants including Alpha, Beta, Delta, Gamma and Omicron based on definition from WHO (https://www.who.int/en/activities/tracking-SARS-CoV-2-variants/). Wild variant refers to wild-type strain—the initial epidemic strain before the Alpha (Wu Y, Kang L, Guo Z, Liu J, Liu M, Liang W. Incubation Period of COVID-19 Caused by Unique SARS-CoV-2 Strains: A Systematic Review and Meta-analysis [published correction appears in JAMA Netw Open. 2022 Sep 1;5(9):e2235424]. JAMA Netw Open. 2022;5(8):e2228008. Published 2022 Aug 1. doi:10.1001/jamanetworkopen.2022.28008), in order to make this clearer, we have corrected “Wild variant” as “wild-type strain”. In addition, we added information in methods as shown in page line:” In this study, we included studies which reported long COVID-19 caused by wild-type strain and variants of concern (VOC) including Alpha, Beta, Delta, Gamma and Omicron in terms of WHO definitions”. Further, except SARS-CoV-2 strains reported by article, we supplemented variants information of the other articles based on genomic epidemiology of SARS-CoV-2 in Global Initiative of Sharing All Influenza Data (GISAID) platform (https://gisaid.org/phylodynamics/global/nextstrain/) to find the main epidemic variant in specific country and period. This platform as an open-source project with pathogen genome data, provide a continually-updated view on SARS-CoV-2 genomes shared from global scientists.

121 I suggest to write that heterogeneity is taken for granted

Response: Thanks. We have added more information on heterogeneity as shown in page 4 line 153-155: “Considering the heterogeneity larger than 50% between studies, random-effects models [19] were used to calculate the pooled effect and its 95% confidence interval (CI)”

Figure 2 Not scientifically appropriate, to remove ..

Response: Thanks. We have removed Figure 2.

Reviewer 2 Report

Dear Authors, the manuscript may have different insights for literature. However, it is necessary to file my concerns:

With “different variants of concern:” what is meant? I recommend being clear in the title

Methodologically I recommend “sketching” the PICO in a discursive way. Readers understand the population of interest, but what outcome did you think of investigating? I also recommend putting the PROSPERO code in the abstract to increase the quality of your work

The conclusions of the abstract are vague and I suggest providing some frank implications on the various symptoms. Possibly also suggest that there is no difference between the different variants

On line 63, it is necessary to provide a rationale that distinguishes your work from that of the other authors you have correctly cited. Did any of them mention Long Syndromes?

In the methods which outcomes have you included and / or excluded?

Figure 2 suggest to remodel it from a scientific point of view

Author Response

Reviewer: 2 Dear Authors, the manuscript may have different insights for literature. However, it is necessary to file my concerns: With “different variants of concern:” what is meant? I recommend being clear in the title. Response: Thanks. In our study, we included wild-type strain, and variants including Alpha, Beta, Delta, Gamma and Omicron based on definition from WHO (https://www.who.int/en/activities/tracking-SARS-CoV-2-variants/). Wild variant refers to wild-type strain—the initial epidemic strain before the Alpha (Wu Y, Kang L, Guo Z, Liu J, Liu M, Liang W. Incubation Period of COVID-19 Caused by Unique SARS-CoV-2 Strains: A Systematic Review and Meta-analysis [published correction appears in JAMA Netw Open. 2022 Sep 1;5(9):e2235424]. JAMA Netw Open. 2022;5(8):e2228008. Published 2022 Aug 1. doi:10.1001/jamanetworkopen.2022.28008), in order to make this clearer, we have corrected “Wild variant” as “wild-type strain”. We have changed title “Comparison of long-term consequences of COVID-19 caused by different variants of concern: a systematic review and me-ta-analysis” as “Comparison of long COVID-19 caused by different SARS-CoV-2 strains: a systematic review and meta-analysis”. Methodologically I recommend “sketching” the PICO in a discursive way. Readers understand the population of interest, but what outcome did you think of investigating? I also recommend putting the PROSPERO code in the abstract to increase the quality of your work Response: Thanks. We have described it using PICO in abstract as shown in page 1 line 23-26: “We systematically searched PubMed, EMBASE, and ScienceDirect to find cohort studies of long COVID-19 defined by WHO. The main outcomes were the percentages of long COVID-19 among patients infected with different SARS CoV-2 strains.” and added it in methods as shown in page 3 line 102-109: “The following PICO model was used to evaluate the study eligibility: (P) Participants: long COVID-19 patients; (I) Intervention: different SARS-CoV-2 strains; (C) Comparator: not applicable; (O) Outcome measure: long COVID-19 including clinical features (general symptoms, respiratory symptoms, cardiovascular symptoms, gastrointestinal symptoms, neurological and psychiatric symptoms), pulmonary functional test (PFT), chest computerized tomography (CT]) and quality of life.” We have added PROSPERO in the abstract as shown in page 1 line 26-27:” This study was registered in PROSPERO (CRD42022339964).” The conclusions of the abstract are vague and I suggest providing some frank implications on the various symptoms. Possibly also suggest that there is no difference between the different variants. Response: Thanks. We have restated conclusions of the abstract as shown in page 1 line 41-45:” The findings of this study suggest that there is no significant difference on long COVID-19 caused by different strains, except some general symptoms and sleep difficulty.”. On line 63, it is necessary to provide a rationale that distinguishes your work from that of the other authors you have correctly cited. Did any of them mention Long Syndromes? Response: Thanks. the cited references discussed the severity of COVID-19, not long COVID-19 which was defined by WHO (Long COVID-19, also called post COVID-19 condition, was defined as condition occurs in individuals with a history of probable or confirmed severe acute respiratory syn-drome coronavirus 2 (SARS CoV-2) infection, usually 3 months from the onset of COVID-19 with symptoms and that last for at least 2 months and cannot be explained by an alternative diagnosis). We have restated introduction to make it clear. In the methods which outcomes have you included and / or excluded? Response: Thanks. We included all outcomes related to Long COVID-19 based on primary references (Long COVID-19, also called post COVID-19 condition, was defined as condition occurs in individuals with a history of probable or confirmed severe acute respiratory syn-drome coronavirus 2 (SARS CoV-2) infection, usually 3 months from the onset of COVID-19 with symptoms and that last for at least 2 months and cannot be explained by an alternative diagnosis). According to suggestions from reviewer 1 and 2, we have described it using PICO it in methods as shown in page 3 line 102-109: “The following PICO model was used to evaluate the study eligibility: (P) Participants: long COVID-19 patients; (I) Intervention: different SARS-CoV-2 strains; (C) Comparator: not applicable; (O) Outcome measure: long COVID-19 including clinical features (general symptoms, respiratory symptoms, cardiovascular symptoms, gastrointestinal symptoms, neurological and psychiatric symptoms), pulmonary functional test (PFT), chest computerized tomography (CT]) and quality of life.” Figure 2 suggest to remodel it from a scientific point of view Response: Thanks. We have removed Figure 2 according to suggestions from reviewer 1.

Round 2

Reviewer 1 Report

Dear Authors, thank you for addressing my concerns, I can suggest the suitability of the paper for publication

Author Response

Dear Editors and Reviewers:

Thank you for your letter and for the reviewers’ comments concerning our manuscript entitled “Comparison of long COVID-19 caused by different SARS-CoV-2 strains: a systematic review and meta-analysis” (Submission ID ijerph-1961520). Those comments are all valuable and very helpful for revising and improving our paper. We have made the requested changes which we hope meets your approval. The changes were noted in *Track Changes* in the revised version. The main modification in the paper and the responds to the reviewers and editors’ comments are as following:

Reviewer(s)' Comments to Author:

Reviewer: 1

Dear Authors, thank you for addressing my concerns, I can suggest the suitability of the paper for publication

Response: Thanks.

Reviewer 2 Report

Dear authors, your review has been thorough, but I would still like to suggest some minor recommendations:

L17 This systematic review and meta-analysis aimed to evaluate ..

L34 What about a particular strain?

L189 by convention, the discussion begins with a paraphrase of the objective, then a summary of the results obtained from your study is subsequently made...

Thanks and Best Regards

Author Response

Reviewer: 2

Dear authors, your review has been thorough, but I would still like to suggest some minor recommendations:

Response: Thanks.

L17 This systematic review and meta-analysis aimed to evaluate ..

Response: Thanks. We have changed “We did a systematic review and meta-analysis to evaluate...” as “This systematic review and meta-analysis aimed to evaluate” as shown in page 1 line 18.

L34 What about a particular strain?

Response: Thanks. We have changed “The findings of this study suggest that there is no significant difference on long COVID-19 caused by different strains, except some general symptoms and sleep difficulty.” as “The findings of this study suggest that there is no significant difference on long COVID-19 caused by different strains, except some general symptoms (Alpha or Omicron variant) and sleep difficulty (wild-type strain)” as shown in page 1 line 34-35

L189 by convention, the discussion begins with a paraphrase of the objective, then a summary of the results obtained from your study is subsequently made...

Response: Thanks. We have added the information in discussion “The emerging unique variants of SARS-CoV-2 indicate that the long COVID-19 may be different due to changes in virulence of variants. This systematic review and meta-analysis comprehensively assessed the long COVID-19 caused by different SARS-CoV-2 strains to supplement limited evidence. Generally, this study reported the most common long COVID-19 and its differences between unique strains.” As shown in page 7-8 line 189-193.